# Spatio-Temporal Variation Characteristics and Driving Forces of Farmland Shrinkage in Four Metropolises in East Asia

**Yaxin Shi [1] and Yishao Shi [2],***

[1]  School of City and Environment, Yunnan University of Finance and Economics, Kunming 650221, Yunnan, China; shiyaxin52@outlook.com
[2]  College of Surveying and Geo-Informatics, Tongji University, Shanghai 200092, China
*  Correspondence: shiyishao@tongji.edu.cn

**Abstract:** The shrinkage of cultivated land is a general trend in metropolitan areas. However, previous studies have mainly paid attention to the shrinkage of arable land in major grain-producing areas, mixed agro-pastoral areas, ecologically fragile areas and construction areas of major engineering projects (such as the Three Gorges project). This paper analyses the characteristics and driving factors of cultivated land change on the metropolitan area scale and longer time dimension. Exemplified by four metropolises in East Asia, Tokyo, Beijing, Shanghai and Guangzhou, based on official statistics for the cities involved, using correlation analysis, principal component analysis and regression analysis methods and Statistical Product and Service Solutions (SPSS) 25 software, the main driving factors and differences in cultivated land shrinkage in Beijing, Shanghai, Guangzhou and the Tokyo metropolitan area are quantitatively revealed. The results show the following: (1) there are some differences in the shrinkage in arable land and spatial distributions among different cities. Tokyo and Guangzhou still have some cultivated land in central urban areas, while there is no arable land in the central areas of Beijing and Shanghai. (2) There is a clear difference in the main driving factors of cultivated land shrinkage between the Tokyo metropolitan area and Beijing, Shanghai and Guangzhou. Residential area and population are the main driving factors for the former, while economic development and urbanisation are the main driving factors for the latter three areas. It shows that the shrinkage of cultivated land is closely related to the developmental stage of urbanisation. (3) There is a rather obvious difference in the main driving factors of cultivated land change among Beijing, Shanghai and Guangzhou. Gross Domestic Product (GDP) is the primary factor leading to the shrinkage of arable land in Beijing, built-up area is the primary factor in Shanghai, and the Engel's coefficient for rural residents is the primary factor in Guangzhou. This reflects the difference in measures for the utilisation and protection of cultivated land among different cities. (4) The socioeconomic factors that affect the contraction of cultivated land are varied. In this study, industrial restructuring is included in the evaluation index system, mainly because industrial transformation and upgrading is essential for sustainable development of emerging global cities, and agricultural production conditions are not included in the evaluation index system, mainly because they are more the result of urbanisation than the cause.

**Keywords:** farmland sustainability; spatio-temporal variation; driving forces; urban agricultural revitalisation; four metropolises of East Asia

## 1. Introduction

Cultivated land resources and their production capacities are related to human food security, ecological security and social and political security [1–5]. How to make better use of and protect

the increasingly valuable cultivated land resources has been one of the focal points of government, academia and the public [6–8], including spatial and temporal changes and the effects of farmland expansion, conversion, abandonment and loss [9–17], influencing factors or driving forces [18–21] and farmland improvement measures and protection strategies [22]. Liu et al. [11] pointed out that obvious losses have consistently emerged in cities with high administrative levels and large population sizes in China since the 1970s. Based on a comprehensive statistical analysis of 70 metropolitan areas in mainland China, Huang et al. [23] investigated regional differences in the trends in grain self-sufficiency capacity in these areas from 1990 to 2015. They found a declining trend in 3/4 of metropolitan areas, mainly located in the rapidly urbanising eastern coastal areas and in the West. Since the beginning of the 21st century, under the background of promoting high-quality urbanisation and strengthening the protection of cultivated land, the shrinkage of cultivated land in metropolitan areas has continued, even in developed countries where the level of urbanisation is already high. Does this imply some kind of rationality? Does it also indicate that current cultivated land protection, with food production capacity at its core, is facing a general policy failure dilemma in metropolitan areas [24]?

In the past ten years, in contrast to the intense research on shrinking cities (i.e., cities which experience demographic and economic decline) [25–31], the phenomenon of farmland shrinkage in metropolitan areas has largely been ignored [11,23]. Therefore, an accurate understanding and grasp of the spatial and temporal evolution of cultivated land in metropolitan areas, the sustainable development of metropolitan areas and the versatility of cultivated land resources is required. A balanced relationship among ecological land preservation, living land protection, cultivated land protection and industrial land guarantee in metropolitan areas should be coordinated as a whole to provide references for the rational allocation and sustainable utilisation of land resources.

East Asia, including China, Japan, South Korea, North Korea and Mongolia, with a land area of 12.5 million square kilometres and a population of 1.6 billion, is one of the most densely populated areas in the world. East Asia is also one of the major regions of world agricultural production. The rice produced accounts for more than 40% of the world's total rice production, tea accounts for more than 25% of the world's total production and soybeans account for more than 20% [32]. The output of cotton, peanut, corn, sugar cane, sesame, rapeseed and silk occupies an important position in the world. The main cities are Beijing, Shanghai, Hong Kong, Tokyo, Seoul, Pyongyang and Guangzhou. This paper selects the four metropolises of the Tokyo Metropolitan Area, Beijing, Shanghai and Guangzhou to compare and analyse the spatial and temporal characteristics of cultivated land contraction.

The basic characteristics of the four metropolises in East Asia are shown in Table 1. Tokyo is the political, economic and cultural centre of Japan, the hub of land, sea and air transportation in Japan, and a modern international metropolis. The Tokyo metropolitan area consists of Tokyo, Saitama, Kanagawa and Chiba Prefectures (Figure 1). In 2017, the total land area was 13,376 square kilometres, and the total population was approximately 35.7 million [33]. Beijing is China's political and cultural centre (Figure 2). It is a world-famous ancient capital and a modern international metropolis. In 2017, the total land area of the city was 16,410 square kilometres, and the resident population was 21.71 million [34]. Shanghai is a rising global city, an international economic, financial, trade, shipping, scientific and technological innovation centre, and a cultural metropolis (Figure 3). In 2017, the administrative area of Shanghai was 6833 square kilometres, with a permanent resident population of 24.1833 million [35]. Guangzhou is the capital of Guangdong province, a famous national, historical and cultural city, an important central city of China, an international trade centre and a comprehensive transportation hub (Figure 4). In 2017, the land area of the city was 7434.40 square kilometres, with a permanent resident population of 14.4984 million [36].

**Table 1.** Four metropolises in East Asia.

| City or Metropolitan Circle | Administrative Areas | Land Area (km²) | Total Population (10,000 People) | Population Density (Person/km²) |
| --- | --- | --- | --- | --- |
| The Tokyo metropolitan area | Including Tokyo, Saitama, Kanagawa and Chiba prefectures | 13,376 | 3570 | 2669 |
| Beijing | Including 16 districts, i.e., Dongcheng, Xicheng, Chaoyang, Shijingshan, Fengtai, Haidian, Fangshan, Tongzhou, Shunyi, Changping, Mentougou, Daxing, Huairou, Pinggu, Miyun and Yanqing | 16,410 | 2170.70 | 1323 |
| Shanghai | Including 16 districts, i.e., Huangpu, Jing 'an, Xuhui, Changning, Yangpu, Hongkou, Putuo, Pudong new area, Baoshan, Jiading, Minhang, Songjiang, Qingpu, Fengxian, Jinshan and Chongming | 6833 (Excluding sea area) | 2418.33 | 3539 |
| Guangzhou | Including 11 districts, i.e., Yuexiu, Haizhu, Liwan, Tianhe, Huangpu, Baiyun, Panyu, Huadu, Nansha, Zengcheng and Conghua | 7434.40 | 1449.84 | 1950 |

Data sources: ① 2019 Japan Statistical Yearbook [33], ② 2018 Beijing Statistical Yearbook [34], ③ 2018 Shanghai Statistical Yearbook [35] and ④ 2018 Guangzhou Statistical Yearbook [36].

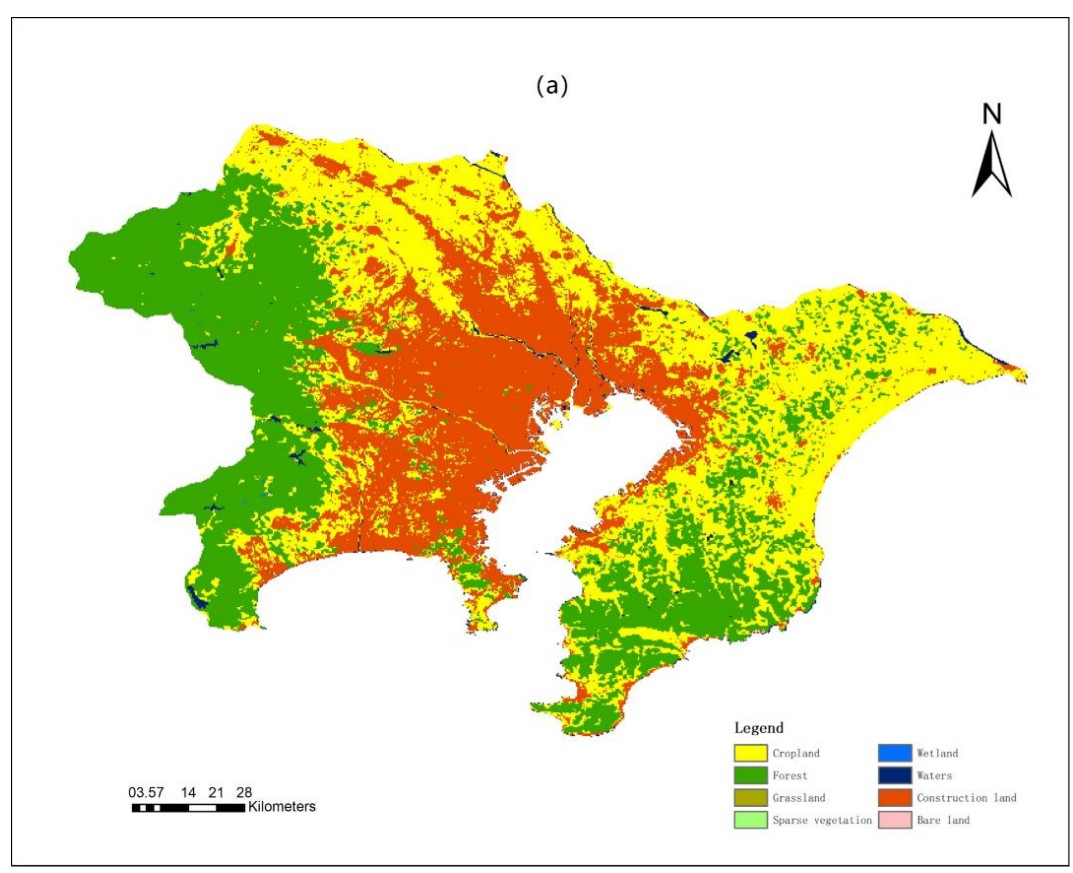

**Figure 1.** *Cont.*

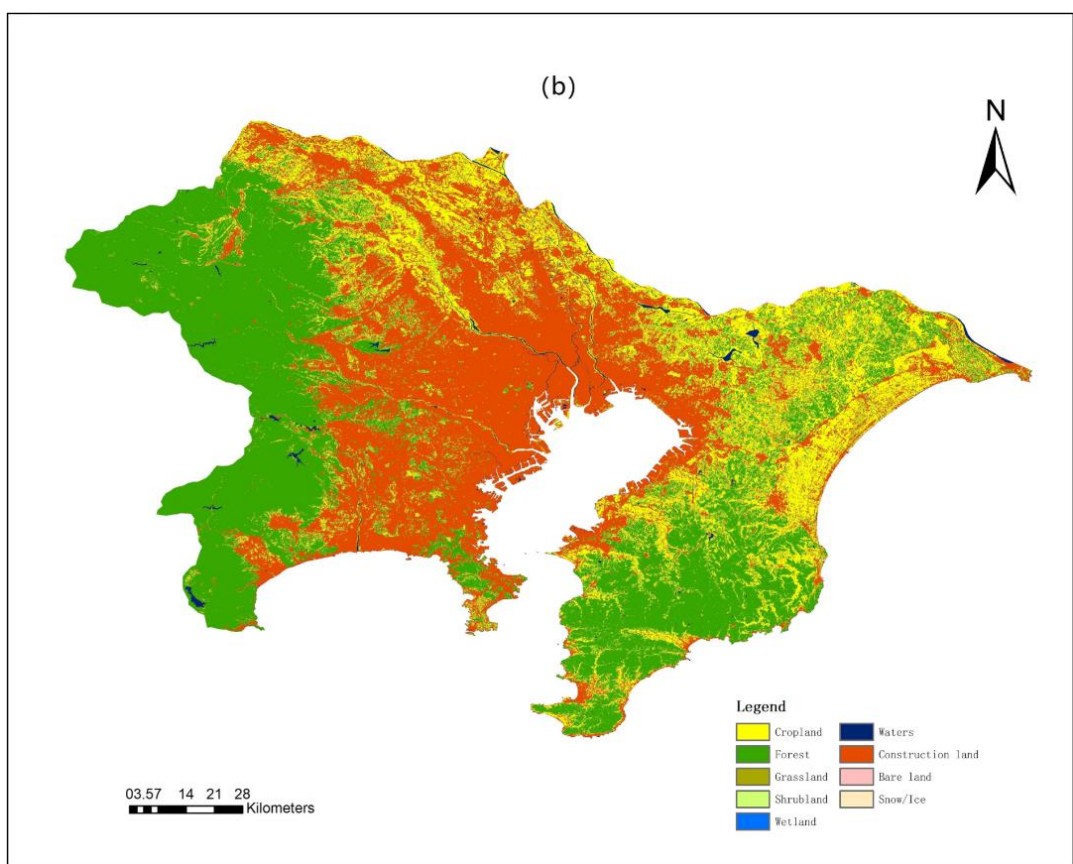

**Figure 1.** Land use map of the Tokyo metropolitan area in (**a**) 2000 and (**b**) 2017. Data source: Resources and Environment Data Cloud Platform. Institute of Geographic Sciences and Resources, Chinese Academy of Sciences (CAS); Data Centre for Resources and Environmental Sciences, CAS.

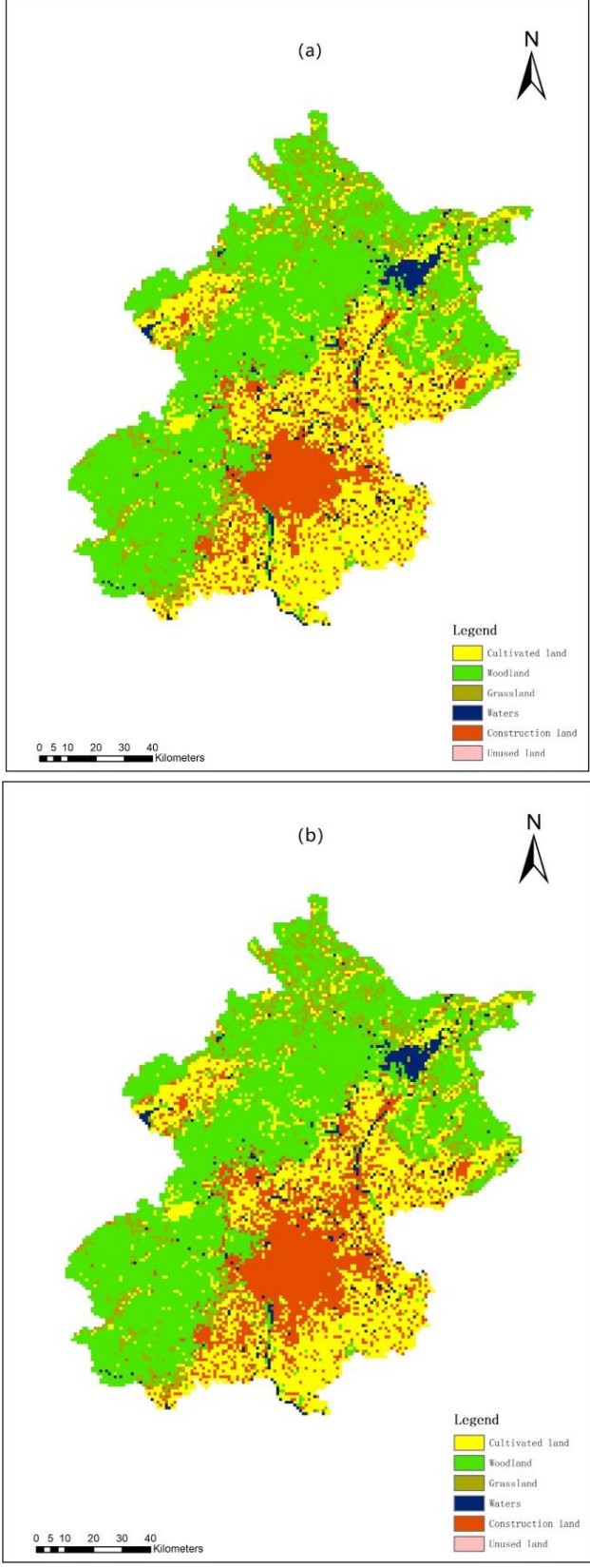

**Figure 2.** *Cont.*

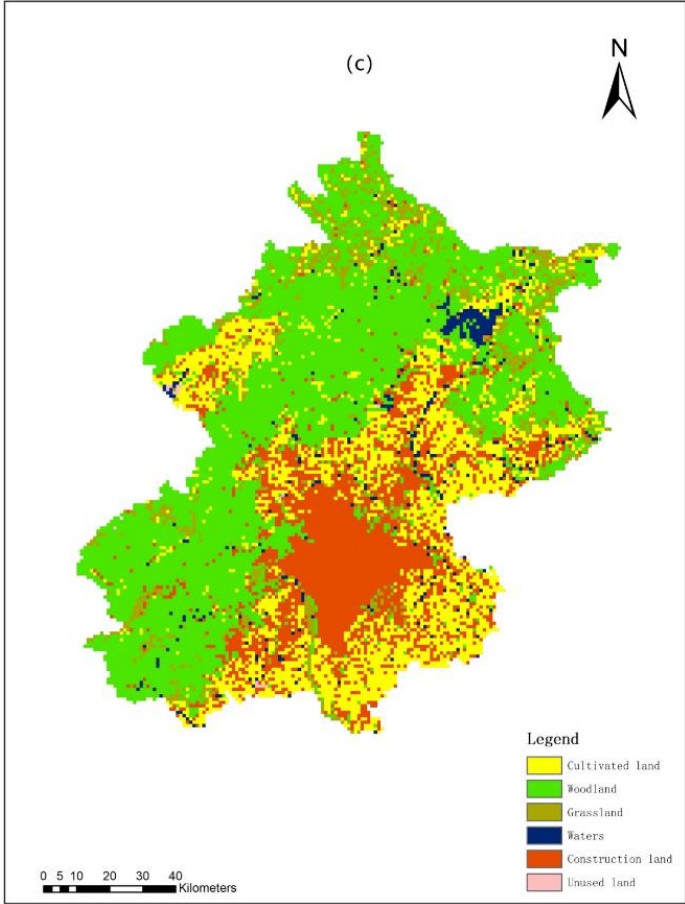

**Figure 2.** Land use map of Beijing in (**a**) 2000, (**b**) 2010 and (**c**) 2018. Data source: Resources and Environment Data Cloud Platform. Institute of Geographic Sciences and Resources, CAS; Data Centre for Resources and Environmental Sciences, CAS.

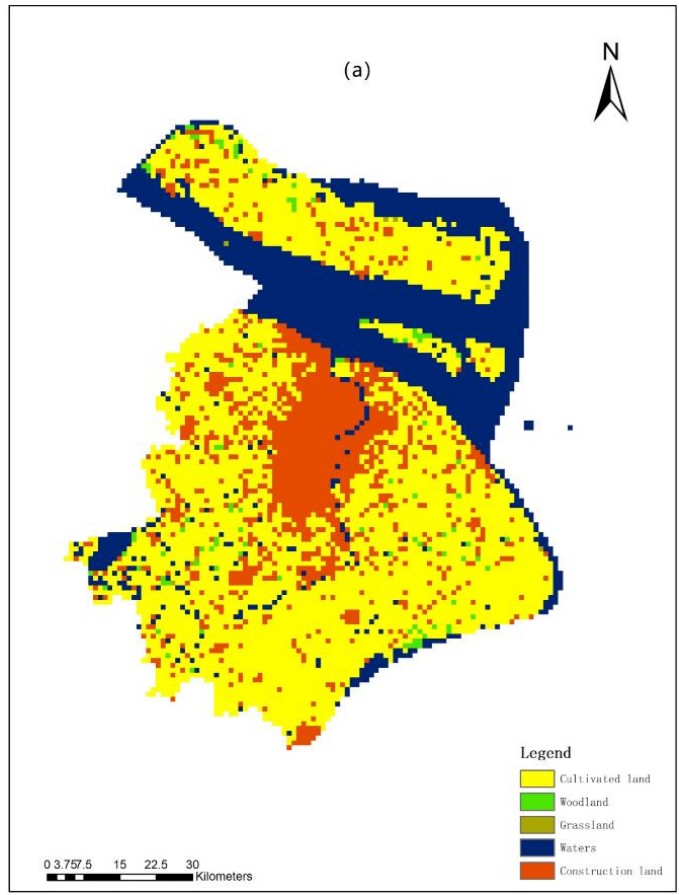

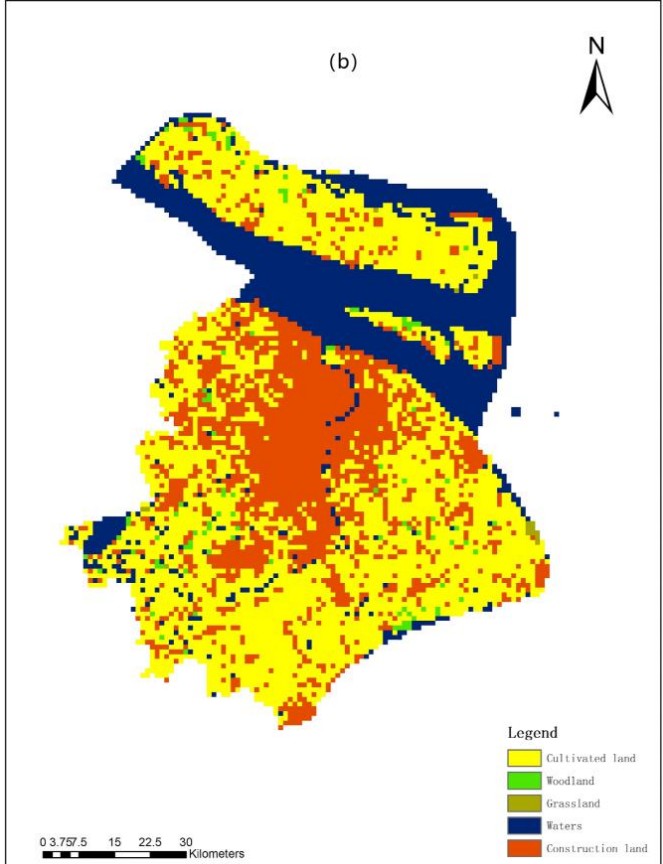

**Figure 3.** *Cont.*

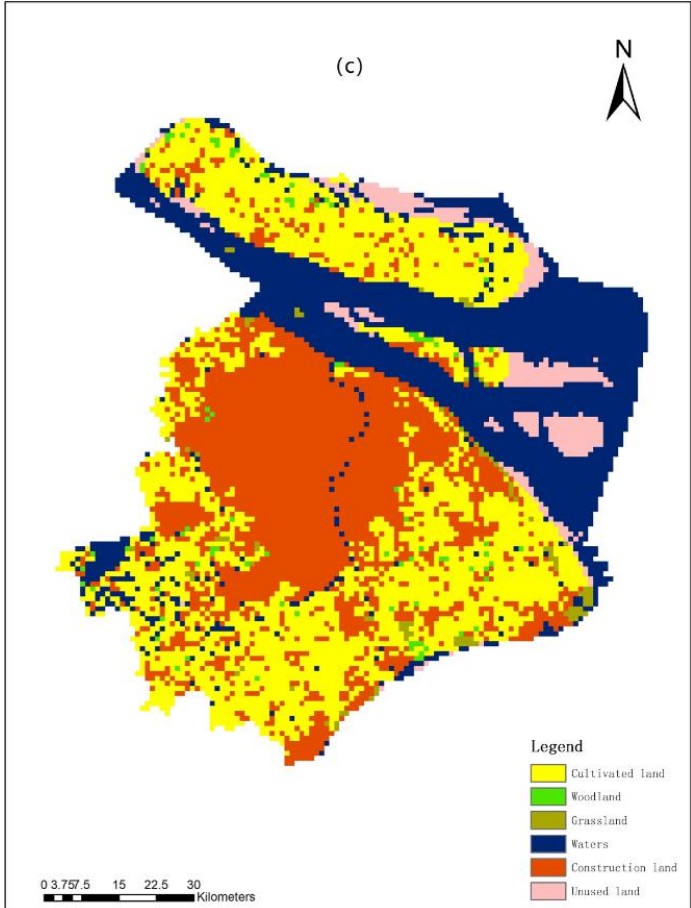

**Figure 3.** Land use map of Shanghai in (**a**) 2000, (**b**) 2010 and (**c**) 2018. Data source: Resources and Environment Data Cloud Platform. Institute of Geographic Sciences and Resources, CAS; Data Centre for Resources and Environmental Sciences, CAS.

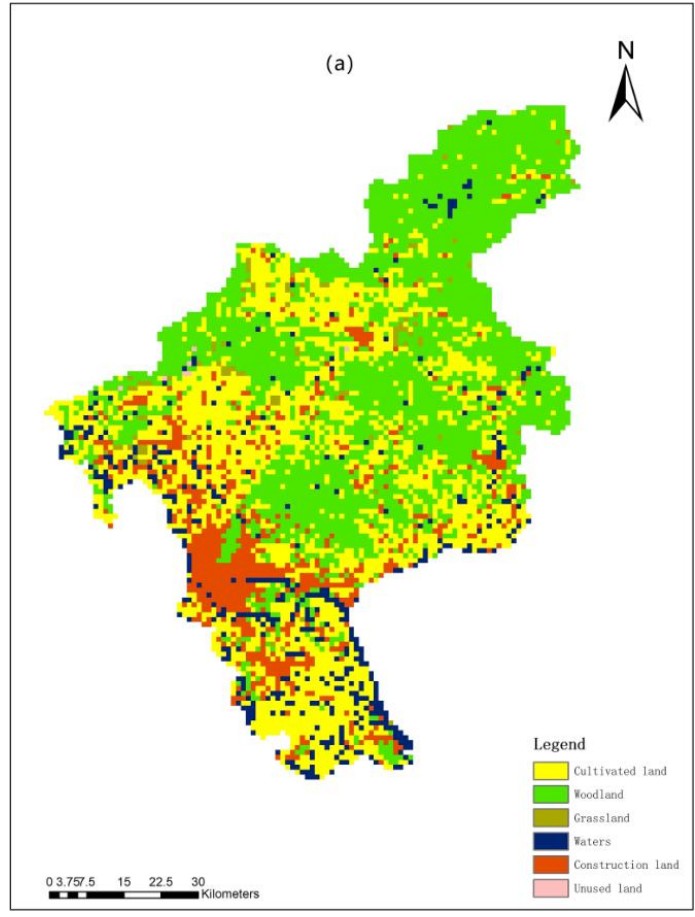

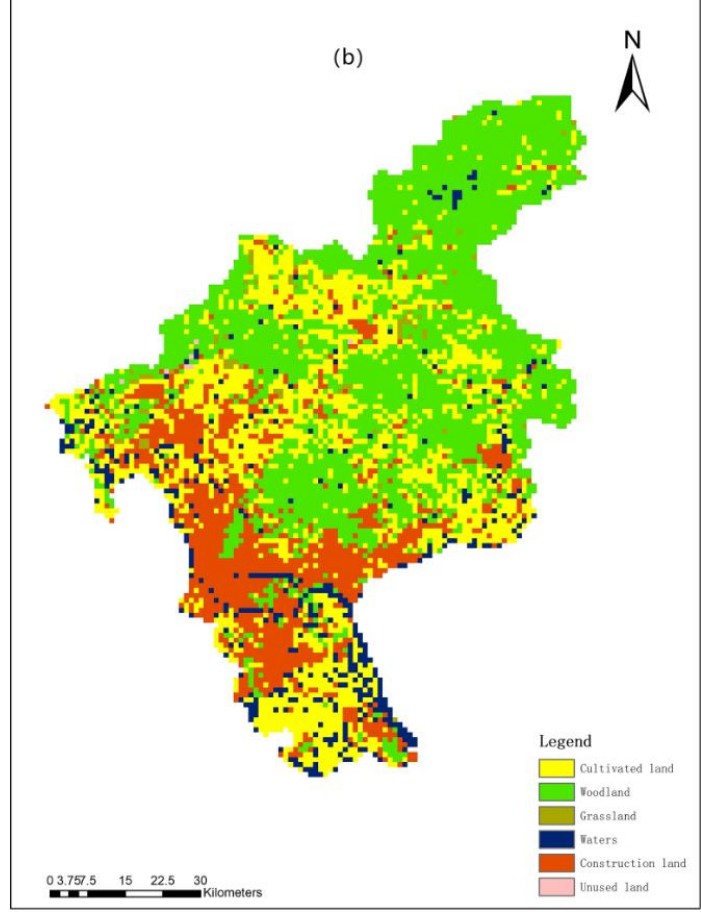

**Figure 4.** *Cont.*

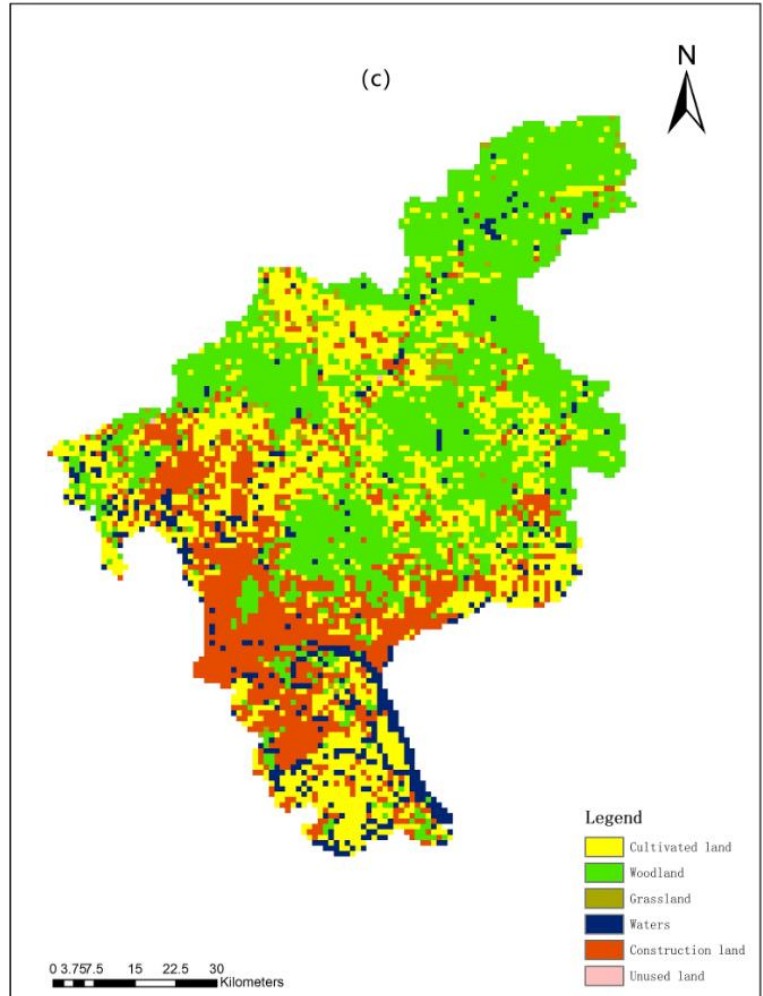

**Figure 4.** Land use map of Guangzhou in (**a**) 2000, (**b**) 2010 and (**c**) 2018. Data source: Resources and Environment Data Cloud Platform. Institute of Geographic Sciences and Resources, CAS; Data Centre for Resources and Environmental Sciences, CAS.

## 2. Data and Methods

### 2.1. Data Sources

In the present study, the data for the Tokyo metropolitan area were from Japan Statistical Yearbooks, Japanese agriculture, forestry and fisheries statistical information, Tokyo Statistical Yearbooks and official statistical websites of Tokyo. The data for Chinese cities were from Beijing Statistical Yearbooks, Shanghai Statistical Yearbooks, Guangzhou Statistical Yearbooks, Statistical Yearbooks of China's Urban Construction and official statistical websites of Beijing, Shanghai and Guangzhou cities.

### 2.2. Index System for Driving Force Analysis

Population growth, economic development (mainly industrialisation) and urbanisation were usually regarded as the three main drivers of urban expansion and shrinking agricultural land (or arable land) [8,10,11,15,18,19,37–41]. Some scholars also investigated the effect of natural factors (e.g., water system, topographic position index, soil and climatic factors) on the change of cultivated land [18,42]. Some scholars measured economic development in detail from three aspects: industrial structure, investment and consumption level and residents' living standard [39]. Other scholars regarded agricultural production conditions or agricultural science and technology as driving factors for the change of cultivated land [38–40].

All these showed that socioeconomic factors were the main driving force for the change of farmland. Moreover, in metropolitan areas, population and economic growth are fast, urbanisation is strong, industrial structure transformation and upgrading are rapid and agricultural production conditions are easy to be improved quickly. Therefore, this paper mainly analyses the socioeconomic driving factors of the change of cultivated land in East Asian metropolises.

It is important to note that owing to certain differences between Japan and China's statistical indicators, this paper fails to build a unified index system for driving factor analysis and could only conduct quantitative analyses separately.

2.2.1. Index System for Driving Force Analysis of Beijing, Shanghai and Guangzhou

This paper selected 16 driving factors (second grade indexes) from four aspects (first grade indexes), including population growth, economic development, industrial restructuring and urbanisation, to build an index system for a driving force analysis (Table 2).

**Table 2.** Pre-selected index system for the driving force analysis of Beijing, Shanghai and Guangzhou.

| Grade I Indexes | Grade II Indexes | Unit |
|---|---|---|
| Population growth (X1) | Resident population (X11) | 10,000 people |
| | Non-agricultural population (X12) | 10,000 people |
| | Resident population density (X13) | person/square kilometer |
| Economic development (X2) | Gross Domestic Product (GDP) (X21) | 100 million yuan |
| | GDP per capita (X22) | yuan |
| | Fixed assets investment (X23) | 100 million yuan |
| | CPI(X24) | |
| | Commodity retail price index (X25) | |
| | Passenger traffic (X26) | 10,000 people |
| | Freight volume(X27) | 10,000 tons |
| Industrial restructuring (X3) | Proportion of output value of primary industry (X31) | % |
| | Proportion of output value of the secondary and tertiary industries (X32) | % |
| Urbanisation (X4) | Proportion of non-agricultural population (X41) | % |
| | Built-up area (X42) | square kilometers |
| | The Engel's coefficient for rural residents (X43) | % |
| | Per capita living area for urban residents (X44) | m$^2$ |

2.2.2. Index System for Driving Force Analysis of the Tokyo Metropolitan Area

This paper selected 8 driving factors from three aspects, including population growth, economic development and people's life to build an index system for a driving force analysis in the Tokyo metropolitan area (Table 3).

**Table 3.** Pre-selected index system for a driving force analysis of the Tokyo metropolitan area.

| Grade I Indexes | Grade II Indexes | Unit |
|---|---|---|
| Population growth | Total population (X11) | 10,000 people |
| | Population density (X12) | person/km$^2$ |
| Economic development | GDP (X21) | 100 million yen |
| | Per capita regional income (X22) | 1000 yen |
| | Investment cost (X23) | millions of yen |
| | Standard land price (X24) | yen/square meter |
| People's life | The Engel's coefficient (X41) | % |
| | Residential area (X42) | square kilometers |

*2.3. Methods*

The research methods of cultivated land change include qualitative and quantitative analysis. The specific quantitative methods include factor analysis, principal component analysis, multiple linear regression, canonical correlation analysis, the historical tracing method and the model analysis method [42]. By combining qualitative and quantitative methods (statistical analysis), this paper analysed the spatial and temporal variation characteristics of cultivated land quantity and distribution in four metropolises.

The research of Su and Cao showed that: for the research of driving change of cultivated land utilisation, the method combined with the principal component analysis had a high frequency of application and was basically applicable to all spatial scales [37]. This method could simplify the complex system, improve the accuracy and was beneficial to determine the main driving factors. Therefore, correlation analysis, principal component analysis and regression analysis were used to select the driving factors of cultivated land change and determine the influencing degree.

2.3.1. Driver Analysis Methods for Beijing, Shanghai and Guangzhou

Correlation Analysis

Pearson correlation analysis was conducted using Statistical Product and Service Solutions (SPSS) 25 software (International Business Machines Corporation 2017), and the correlation coefficients of each driving factor and cultivated land area were obtained. If the threshold of the correlation coefficient was set as 0.4, a driving force index whose correlation coefficient was less than the threshold should be removed. The correlation coefficients of X24 and X27 were below 0.4; thus, they were removed.

Principal Component Analysis (PCA)

In order to eliminate the influence of different dimensions, the original data were standardised. Then, Kaiser-Meyer-Olkin (KMO) and Bartlett spherical tests were conducted to test whether the index data were suitable for PCA. As seen in Table 4, the observed Bartlett test statistic was 842.915, the degree of freedom was 91, and the probability was close to 0, less than the significance level of 0.05, the KMO value was 0.663 > 0.6, indicating that it was suitable for factor analysis.

**Table 4.** Kaiser-Meyer-Olkin (KMO) and Bartlett spherical tests in Beijing.

| Kaiser-Meyer-Olkin Measure of Sampling Adequacy | | 0.663 |
|---|---|---|
| Bartlett test for sphericity | Approximate Chi-Square | 842.915 |
| | Degrees of freedom (df) | 91 |
| | Statistical significance (Sig.) | 0.000 |

In Beijing, the PCA results showed that the cumulative contribution rate of the first and second principal components reached 91.599% (Table 5). Therefore, most of the information of the factor could be generalised by extracting only two principal components.

**Table 5.** The results of principal component analysis (PCA) in Beijing.

| Component | Total | % of Variance | Cumulative % |
|---|---|---|---|
| 1 | 11.572 | 82.654 | 82.654 |
| 2 | 1.252 | 8.945 | 91.599 |
| 3 | 0.646 | 4.614 | 96.213 |
| 4 | 0.380 | 2.712 | 98.925 |
| 5 | 0.073 | 0.518 | 99.443 |
| 6 | 0.042 | 0.303 | 99.746 |
| 7 | 0.022 | 0.159 | 99.905 |
| 8 | 0.008 | 0.059 | 99.964 |
| 9 | 0.003 | 0.024 | 99.988 |
| 10 | 0.001 | 0.009 | 99.997 |
| 11 | 0.000 | 0.003 | 99.999 |
| 12 | 0.000 | 0.001 | 100.000 |
| 13 | 0.000 | 0.000 | 100.000 |
| 14 | 0.000 | 0.000 | 100.000 |

Factors were then rotated by Varimax, and factor loadings were calculated. As seen in Table 6, $X_{11}$, $X_{12}$, $X_{13}$, $X_{22}$, $X_{41}$ and $X_{44}$ had higher loads on the first principal component, and $X_{25}$ and $X_{26}$ had higher loads on the second principal component. It can be concluded that the driving factors of the first principal component in Beijing are population, economic and urbanisation factors, and the driving factors of the second principal component are commodity retail price index and passenger traffic.

**Table 6.** Factor loading matrix after rotation in Beijing.

| Index | First Principal Component | Second Principal Component |
|---|---|---|
| $X_{11}$ | 0.980 | −0.105 |
| $X_{12}$ | 0.984 | −0.080 |
| $X_{13}$ | 0.983 | −0.092 |
| $X_{21}$ | 0.940 | −0.250 |
| $X_{22}$ | 0.962 | −0.197 |
| $X_{23}$ | 0.955 | −0.236 |
| $X_{25}$ | 0.275 | 0.892 |
| $X_{26}$ | 0.683 | 0.487 |
| $X_{31}$ | −0.955 | −0.073 |
| $X_{32}$ | 0.958 | 0.074 |
| $X_{41}$ | 0.966 | 0.105 |
| $X_{42}$ | 0.895 | 0.067 |
| $X_{43}$ | −0.927 | −0.093 |
| $X_{44}$ | 0.995 | −0.045 |

Similarly, the analysis results of Shanghai and Guangzhou were obtained. Due to the lack of statistical data for some years in Shanghai and Guangzhou, when selecting the above driving force indicators, the SPSS results showed that the matrix was not a positive definite matrix. After eliminating $X_{31}$, the obtained KMO and Bartlett sphericity test values were found to have better effects for factor analysis.

In Shanghai, $X_{11}$, $X_{12}$, $X_{13}$, $X_{22}$, $X_{23}$ and $X_{41}$ had higher loads on the first principal component, and $X_{25}$ and $X_{42}$ had higher loads on the second principal component. It can be concluded that the driving factors of the first principal component are population and economic factors, and the driving factors of the second principal component are commodity retail price index and built-up

area. In Guangzhou, X11, X12, X13 and X22 had higher loads on the first principal component, and X42 and X25 had a higher load on the second principal component. Thus, the driving factors of the first principal component are population and economic factors, and the driving factors of the second principal component are built-up area and commodity retail price index.

Regression Analysis

Through the above analysis, two unrelated principal components (Y1 and Y2) were obtained. Taking Y1 and Y2 as independent variables and the cultivated land area of Beijing (Y) as dependent variables, the regression model between the dependent variable and principal components was established, and the following regression equation was obtained:

$$Y = -2.301Y1 + 1.103Y2 + 31.512 \tag{1}$$

In order to determine the validity of the analysis results, the regression model was tested so that $R^2 = 0.906$ was close to 1, indicating that the regression model had a high degree of fit (Table 7). F-test and *t*-test results were both less than 0.05 (Table 8), indicating that the regression model had statistical significance and that the regression equation could be used to explain the change characteristics of the cultivated land area.

**Table 7.** Goodness of fit of the regression model.

| Model | R | $R^2$ | Adjusted $R^2$ | Errors in Standard Estimates |
|-------|-----|-------|----------------|------------------------------|
| 1 | 0.952 | 0.906 | 0.900 | 2.684 |

**Table 8.** F-test and *t*-test of the regression model.

| F-Test | | *t*-Test | |
|--------|--------------------------------|----------|-----|
| F | Statistical Significance (sig) | t | sig |
| 148.724 | 0.000 | −10.251 | 0.000 |
| | | 1.601 | 0.120 |

To make the model more intuitive, the independent variable was restored to the original variable; that is, the expressions of F1 and F2 regarding the original variable were substituted into the regression Equation (1), and the final regression model was obtained as follows:

$$Y = -0.766X11 - 0.744X12 - 0.756X13 - 0.882X21 - 0.845X22 - 0.879X23 + 0.693X25 + 0.018X26 + 0.574X31 - 0.575X32 - 0.549X41 - 0.540X42 + 0.536X43 - 0.718X44 + 31.512 \tag{2}$$

Similarly, final regression models were obtained for Shanghai and Guangzhou as follows:

$$Y = -0.513X11 - 0.491X12 - 0.515X13 - 0.560X21 - 0.529X22 - 0.510X23 - 0.114X25 - 0.556X26 - 0.305X32 - 0.473X41 - 0.644X42 + 0.266X43 - 0.088X44 + 27.599 \tag{3}$$

$$Y = -0.288X11 - 0.284X12 - 0.180X13 - 0.115X21 - 0.154X22 - 0.121X23 + 0.320X25 - 0.128X26 - 0.361X32 - 0.298X41 - 0.072X42 + 0.365X43 - 0.314X44 + 13.071 \tag{4}$$

### 2.3.2. Driver Analysis Methods for the Tokyo Metropolitan Area

Correlation Analysis

The Pearson correlation analysis showed that the correlations between per capita regional income (X22), investment cost (X23) and standard land price (X24) and the change of cultivated land area were not significant; thus, they were excluded.

Principal Component Analysis (PCA)

To eliminate the influence of different dimensions, the original data were standardised. Then, KMO and Bartlett spherical tests were conducted to test whether the index data were suitable for PCA. As seen in Table 9, the observed Bartlett test statistic was 151.355, the degree of freedom was 10, and the probability was close to 0, less than the significance level of 0.05. The KMO value was 0.807 > 0.7, indicating that it was suitable for factor analysis. The PCA result for the Tokyo metropolitan area is shown in Table 10. The factor loading matrix after rotation for the Tokyo metropolitan area is shown in Table 11.

**Table 9.** KMO and Bartlett spherical tests in the Tokyo metropolitan area.

| Kaiser-Meyer-Olkin Measure of Sampling Adequacy | | 0.807 |
|---|---|---|
| **Bartlett test for sphericity** | Approximate Chi-Square | 151.355 |
| | Degrees of freedom (df) | 10 |
| | Statistical significance (Sig.) | 0.000 |

**Table 10.** The PCA results for the Tokyo metropolitan area.

| Component | Total | % of Variance | Cumulative % |
|---|---|---|---|
| 1 | 3.921 | 78.427 | 78.427 |
| 2 | 0.728 | 14.566 | 92.994 |
| 3 | 0.343 | 6.856 | 99.850 |
| 4 | 0.004 | 0.087 | 99.936 |
| 5 | 0.003 | 0.064 | 100.000 |

**Table 11.** Factor loading matrix after rotation for the Tokyo metropolitan area.

| Index | First Principal Component |
|---|---|
| X11 | 0.979 |
| X12 | 0.980 |
| X21 | 0.608 |
| X31 | 0.812 |
| X32 | 0.986 |

For the Tokyo metropolitan area, X11, X12 and X32 had higher loads on the first principal component. It can be concluded that the driving factors of the first principal component are population and life factors.

Regression Analysis

Through the above analysis, two unrelated principal components (Y1 and Y2) were obtained. Taking Y1 and Y2 as independent variables and the cultivated land area of the Tokyo metropolitan area (Y) as a dependent variable, the regression model between the dependent variable and principal components was established, and the following regression equation was obtained:

$$Y = -1.886Y1 + 25.320 \tag{5}$$

In order to determine the validity of the analysis results, the regression model was tested so that $R^2 = 0.856$ was close to 1, indicating that the degree of fit of the regression model was relatively high (Table 12). The F-test and *t*-test results were both less than 0.05 (Table 13), indicating that the regression model had statistical significance, and the regression equation could be used to explain the change characteristics of the cultivated land area.

**Table 12.** Goodness of fit of the regression model.

| Model | R | $R^2$ | Adjusted $R^2$ | Errors in Standard Estimates |
|---|---|---|---|---|
| 1 | 0.925 | 0.856 | 0.849 | 1.075 |

**Table 13.** F-test and *t*-test of the regression model.

| F-Test | | *t*-Test | |
|---|---|---|---|
| **F** | **Statistical Significance (sig)** | ***t*** | **sig** |
| 124.461 | 0.000 | −11.156 | 0.000 |

To make the model more intuitive, the independent variable was restored to the original variable; that is, the expressions of F1 and F2 regarding the original variable were substituted into the regression Equation (5), and the final regression model was obtained as follows:

$$Y = -0.933X11 - 0.934X12 - 0.579X21 - 0.773X31 - 0.939X32 + 25.320 \tag{6}$$

## 3. Results

### 3.1. Spatio-Temporal Variation in Cultivated Land Shrinkage in Five Metropolises in East Asia

3.1.1. Farmland Shrinkage in the Tokyo Metropolitan Area

In terms of changes in the scale of arable land, (1) the cultivated area in both Tokyo and the Tokyo metropolitan area (i.e., Tokyo and the surrounding three counties) continued to decrease from 1965 to 2015. For example, the average annual net cultivated area of Tokyo decreased by 411.4 hectares, with an average annual decrease rate of 1.48%. The average annual net loss of cultivated land in the Tokyo metropolitan area was 3849.4 hectares, with an average annual decrease rate of 0.91%. (2) As the number of agricultural households has continued to decrease during the process of urbanisation, the average arable land area of each household has generally shown a trend of first decreasing and then increasing (Table 14).

**Table 14.** Changes in cultivated land in Tokyo and the Tokyo metropolitan area from 1965 to 2015.

| Year | Tokyo | | | The Tokyo Metropolitan Area | | |
|---|---|---|---|---|---|---|
| | Cultivated Area ($hm^2$) | Agricultural Population (Households) | The Area of Arable Land Per Household ($hm^2$/Household) | Cultivated Area ($hm^2$) | Agricultural Population (Households) | The Area of Arable Land Per Household ($hm^2$/Household) |
| 1965 | 27,700 | 45,002 | 0.62 | 422,300 | 440,799 | 0.96 |
| 1970 | 18,600 | 38,400 | 0.48 | 372,700 | 406,726 | 0.92 |
| 1975 | 14,600 | 31,019 | 0.47 | 329,700 | 362,888 | 0.91 |
| 1980 | 13,300 | 28,688 | 0.46 | 311,200 | 339,026 | 0.92 |
| 1985 | 12,500 | 26,568 | 0.47 | 297,900 | 316,259 | 0.94 |
| 1990 | 11,500 | 20,679 | 0.56 | 284,400 | 280,236 | 1.01 |
| 1995 | 9980 | 17,367 | 0.57 | 269,980 | 247,915 | 1.09 |
| 2000 | 9000 | 15,460 | 0.58 | 256,200 | 222,533 | 1.15 |
| 2005 | 8340 | 13,700 | 0.61 | 247,540 | 204,636 | 1.21 |
| 2010 | 7670 | 13,099 | 0.59 | 236,670 | 187,768 | 1.26 |
| 2015 | 7130 | 11,222 | 0.64 | 229,830 | 162,588 | 1.41 |

Data source: Japanese agriculture, forestry and fisheries statistical information [43].

The temporal change trend shows that (1) during the rapid growth period of industrialisation and urbanisation (1965–1975) [44], the average annual reduction of arable land area in the Tokyo metropolitan area was 9260 hectares, showing a rapid contraction and the per capita arable land area also tended to decline. (2) During the period of steady development of industrialisation and urbanisation (1975–2000), the average annual reduction in arable land area in the Tokyo metropolitan area was 2940 hectares, showing a medium-speed contraction; however, the per capita arable land area tended to increase. (3) During the later period of industrialisation and urbanisation (after 2000), the average annual reduction in arable land area in the Tokyo metropolitan area was 1758 hectares, showing a low-speed shrinkage and the per capita arable land area continued to increase (Figure 5).

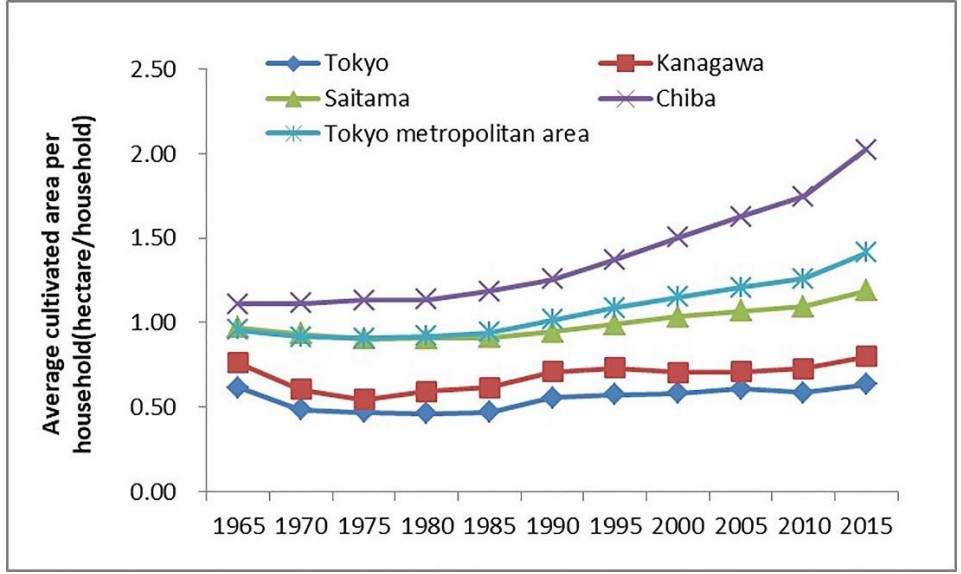

**Figure 5.** Changes in cultivated land in Tokyo and the Tokyo metropolitan area from 1965 to 2015.

From the utilisation rate of arable land (i.e., multiple crop index, which refers to the ratio of sown area to cultivated area) [22,45], the following can be seen: (1) The multiple crop index of the Tokyo metropolitan area tended to decline from 1956 to 1980. There was an increase between 1980 and 1985, but then there was an overall decline. (2) The variation in the multiple crop index of Tokyo went from being generally lower than the surrounding three counties (1956–1980) to slightly higher than the surrounding three counties (1985–2010) and then to generally lower than the surrounding three counties (after 2010). In addition, the multiple crop indices for the Tokyo metropolitan area fell below 100%, and the gap between them gradually narrowed (Figure 6).

In terms of spatial distribution, the proportion of cultivated land in the Tokyo metropolitan area in 2015 was approximately 17.19%, and most of the cultivated land was distributed in the surrounding three counties. In 2015, there were still 7130 hectares of cultivated land in Tokyo, which accounted for only 3.39% of the total area of Tokyo. However, it was mainly distributed in the city department, and the county and island areas were dominated by forests and fields.

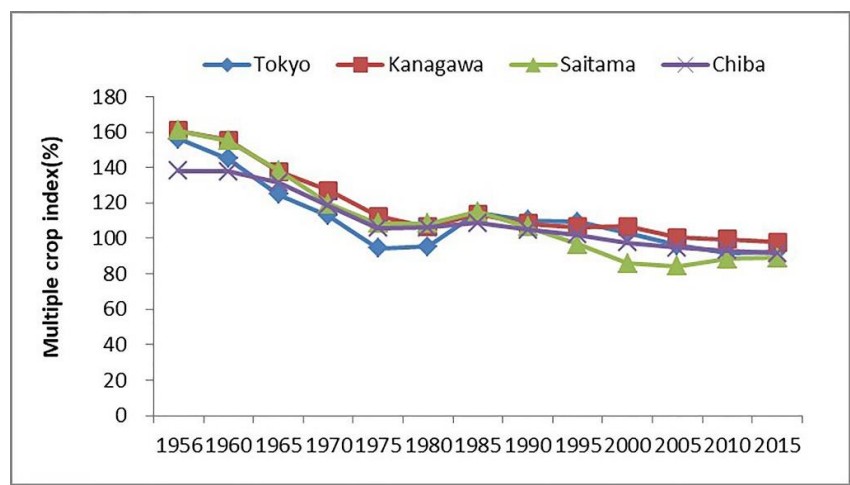

**Figure 6.** Changes in cultivated land utilisation rate in the Tokyo metropolitan area from 1956 to 2015.

3.1.2. Farmland Shrinkage in Beijing City

Since 1980, the area of cultivated land in Beijing has been continuously shrinking. The total scale of cultivated land shrank from 4258 square kilometres in 1980 to 2163 square kilometres in 2016, a decrease of approximately 49% in 36 years (Figure 7). The area of cultivated land owned by the average agricultural population also decreased from 471 square metres to 74 square metres per agricultural population, a decrease of 5.36 times.

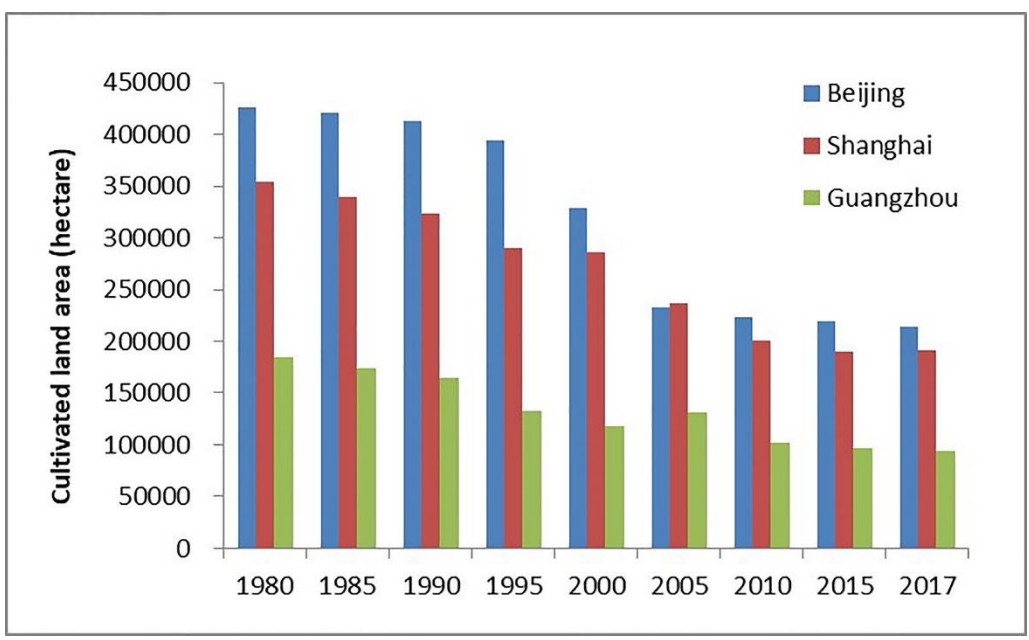

**Figure 7.** Changes in cultivated land area in Beijing, Shanghai and Guangzhou. (Data source: Beijing, Shanghai and Guangzhou Statistical Yearbook [34–36]).

Temporal change trends can be roughly divided into three stages. In Stage I (1980–1994), there was no significant change in the total amount of cultivated land, and there was no large occupation of cultivated land by construction land. In Stage II (1995–2004), the cultivated land area continued to decline, and urban expansion had been realised by occupying cultivated land. In Stage III (2005–2016), the reduction of cultivated land area was controlled. With the awareness of ecological protection and food security, cultivated land protection policies and agricultural land consolidation and transformation were promoted.

In Beijing, the multiple crop index declined continuously from 1980 to 1994, rose on the whole from 1994 to 1999, declined from 2000 to 2003, continued to rise from 2003 to 2010 and dropped significantly again after 2010. In particular, it has dropped below 100% since 2014 (Figure 8). In addition, the utilisation rate of arable land in the southeast was significantly higher than that in the northwest due to land conditions in Beijing, such as topography and fertility.

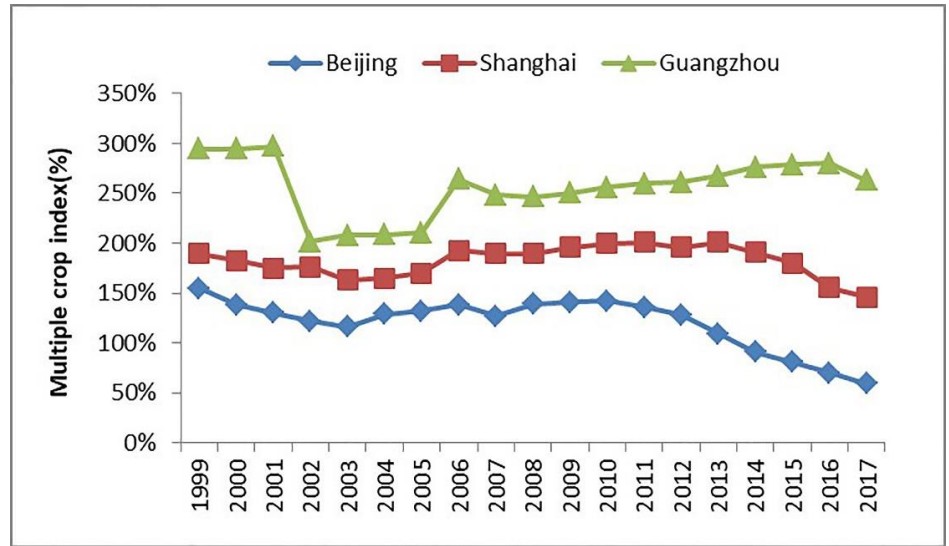

**Figure 8.** Changes in multiple crop indices in Beijing, Shanghai and Guangzhou. (Data source: Beijing, Shanghai and Guangzhou Statistical Yearbook [34–36]).

In terms of spatial distribution, all cultivated land in Beijing is distributed in suburbs, among which the outer suburbs accounted for approximately 97% in 2016, whereas the near suburbs accounted for only 3% (Table 15). However, the arable land area continues to shrink in both the near and outer suburbs.

**Table 15.** The regional distribution of and change in cultivated land in Beijing (Unit: hm$^2$; Data source: Beijing Statistical Yearbook [34]).

| Year | Urban Central Area | Inner Suburbs | Outer Suburbs |
|------|--------------------|---------------|---------------|
| 2007 | 0 | 11,096 | 221,093 |
| 2008 | 0 | 10,791 | 220,897 |
| 2012 | 0 | 7143 | 213,723 |
| 2013 | 0 | 6996 | 214,161 |
| 2014 | 0 | 6800 | 213,148 |
| 2015 | 0 | 6737 | 212,590 |
| 2016 | 0 | 6555 | 209,790 |

Note: In Beijing, the urban central area includes Dongcheng and Xicheng Districts, the inner suburbs include Chaoyang, Shijingshan, Fengtai and Haidian Districts and the outer suburbs include Fangshan, Tongzhou, Shunyi, Changping, Mentougou, Daxing, Huairou and Pinggu Districts, and Miyun and Yanqing Counties.

### 3.1.3. Farmland Shrinkage in Shanghai City

The area of cultivated land in Shanghai has generally contracted since 1978, and it has decreased by approximately 88% in the past 40 years. On average, the cultivated land area per rural population or rural labour force tended to decrease in volatility, but it began to stop falling and increased after 2011 (Figure 7).

The variation in the multiple crop index has also fluctuated. It was stable at approximately 190% in the 1990s, dropped to approximately 160–180% between 2000 and 2005, rose again to approximately

190–200% during 2006–2014, dropped again significantly since 2015 and was only 146% in 2017 (Figure 8).

From the perspective of spatial distribution, the cultivated land in Shanghai was all distributed in the suburbs, mainly in the outer suburbs, and there was no cultivated land in the central urban area. Moreover, the area of arable land continued to shrink in both the inner and outer suburbs.

### 3.1.4. Farmland Shrinkage in Guangzhou City

Since 1978, the arable land area of Guangzhou has been continuously shrinking and has decreased by approximately 1.7 times in the past 40 years. The fluctuation in the arable land owned by the agricultural population has tended to decrease, and the amount of arable land has decreased twice in the past 40 years (Figure 7).

In general, the multiple crop index tended to rise during fluctuation, hovering at approximately 200–220% from 1999 to 2005, dropping to 160–180% from 2006 to 2012, and rising again to 270–285% after 2013 (Figure 8).

From the perspective of spatial distribution, cultivated land in Guangzhou is mainly distributed in the outer suburbs, accounting for approximately 80% of the total cultivated land in the whole city. In spite of fluctuations from 1978 to 2017, it has remained stable overall. The area of arable land in the inner suburbs shrank rapidly, accounting for approximately 19% of the city. The arable land area in the central urban area also shrank rapidly, with 971 hectares of arable land still remaining by 2017, although it accounted for only 1% of the total cultivated land area in the whole city (Table 16).

**Table 16.** The regional distribution and change in cultivated land in Guangzhou (Unit: $hm^2$).

| Year | Urban Central Area | Inner Suburbs | Outer Suburbs |
|---|---|---|---|
| 2002 | 3395 | 63,353 | 73,666 |
| 2003 | 3184 | 54,377 | 78,057 |
| 2004 | 2940 | 51,250 | 78,405 |
| 2005 | 2747 | 43,958 | 84,531 |
| 2007 | 1412 | 34,075 | 70,016 |
| 2008 | 1311 | 33,825 | 68,990 |
| 2009 | 1201 | 32,338 | 68,656 |
| 2010 | 1053 | 32,319 | 68,812 |
| 2011 | 1011 | 32,102 | 68,255 |
| 2012 | 939 | 17,490 | 80,658 |
| 2013 | 919 | 17,436 | 79,793 |
| 2014 | 913 | 17,100 | 78,385 |
| 2015 | 903 | 18,842 | 75,666 |
| 2016 | 1048 | 18,599 | 75,541 |
| 2017 | 971 | 17,541 | 73,596 |

Data source: Guangzhou Statistics Bureau. 2018 Guangzhou Statistical Yearbook. Beijing: China Statistics Press, 2018 [36]. Note: In Guangzhou, the urban central area includes Yuexiu, Haizhu, Liwan and Tianhe Districts, the inner suburbs include Huangpu, Baiyun and Panyu Districts and the outer suburbs include Huadu, Nansha, Zengcheng and Conghua Districts.

### 3.2. Analysis of the Driving Factors of Cultivated Land Shrinkage

### 3.2.1. Driving Factors of Cultivated Land Shrinkage in Beijing, Shanghai and Guangzhou

According to Equation (2), the absolute values of the regression coefficients for each indicator, in descending order, were as follows: GDP (X21), fixed assets investment (X23), GDP per capita (X22), resident population (X11), resident population density (X13), non-agricultural population (X12), per capita living area of urban residents (X44), commodity retail price index (X25), proportion of output

value of the secondary and tertiary industries (X32), proportion of output value of primary industry (X31), proportion of non-agricultural population (X41), built-up area (X42), the Engel's coefficient (i.e., the share of total food expenditure in total personal consumption expenditure) for rural residents (X43) and Passenger traffic (X26). We found that GDP, fixed assets investment, GDP per capita and resident population were the main factors causing the shrinkage of cultivated land in Beijing. Moreover, all four factors had negative effects.

According to Equation (3), we concluded that built-up area, GDP, passenger traffic and GDP per capita were the main factors causing the shrinkage of cultivated land in Shanghai, and they had negative effects.

According to Equation (4), we found that the Engel's coefficient for rural residents, proportion of output value of the secondary and tertiary industries, commodity retail price index and per capita living area of urban residents were the main factors causing the shrinkage of cultivated land in Guangzhou. Among them, commodity retail price index and the Engel's coefficient for rural residents had positive effects, and proportion of output value of the secondary and tertiary industries and per capita living area of urban residents had negative effects.

### 3.2.2. Driving Factors of Cultivated Land Shrinkage in the Tokyo Metropolitan Area

According to Equation (6), we concluded that residential area, population density and total population were the main factors causing the shrinkage of cultivated land in the Tokyo metropolitan area, all of which had negative effects.

## 4. Discussion

Much research has been conducted on cultivated land changes and driving forces based on different spatial scales, such as national, regional, provincial and county scales [42]. However, studies at the regional scale have mostly focused on ecologically fragile areas, mixed agricultural and pastoral areas, major grain-producing areas and areas with major projects, and few studies have been based on mega-cities or metropolitan areas [46]. Therefore, this paper focused on research on the scale of mega-cities or metropolitan areas.

The utilisation rate of cultivated land is an important index reflecting the utilisation degree of cultivated land. With the exception of the Guangzhou metropolitan area, the multiple crop indices for Tokyo, Beijing and Shanghai metropolitan areas generally showed downward trends. On the one hand, the multiple crop indices are related to the natural geographic latitude, farming system and social and cultural customs. On the other hand, resource constraints and environmental pressures, the extent of modern agricultural technology application and promotion, and the development level of local agriculture also affect the variation in multiple crop indices. The decrease in the multiple crop index represents the decrease of the utilisation rate of arable land, but it is not completely the result of the extensive use of arable land. Rather, it may be the result of a variety of factors, such as the diversification of arable land, the existence of agricultural alternatives, the ageing of the population, the occurrence of natural disasters, and the presence of unreasonable food prices. Therefore, it is necessary to pay attention to the opportunity, environmental and social costs of cultivated land, watch the mixed use of cultivated land and enhance the comprehensive productivity and value of cultivated land in metropolitan areas instead of paying attention only to the change in a single index, such as the multiple crop index.

The selection of driving factors is inevitably subjective. However, generally speaking, these mainly include natural factors, economic factors, social factors, technological factors and policy factors [39,42,47]. Some people consider that natural factors are the dominant factors, others think that socioeconomic factors are dominant, and others emphasise a combination of factors [48]. Ye and Dong [49] divided the industrial development into two phases from 1980 to 2002 in Guangzhou, i.e., metaphase of industrialisation (1980–1993) and anaphase of industrialisation (1994–2002), and then revealed quantitative differences in cultivated land change and driving forces. In fact, the driving

factors should vary over time and across regions. Because the time dimension and spatial scale of observation are different, it is normal to obtain a variety of research results. In this study, we mainly consider social factors (population growth and urbanisation) and economic factors (economic development and industrial restructuring), but we do not consider natural factors exhibiting few changes or policy factors that are difficult to quantify.

On the time dimension, the shrinkage of cultivated land is closely related to the developmental stage of industrialisation and urbanisation. Generally, in the early stage of industrialisation and urbanisation, the arable land area in metropolises shrinks slowly. In the middle stage of industrialisation and urbanisation, urban expansion occurs at the cost of rapidly shrinking arable land or agricultural land. In the late stage of industrialisation and urbanisation, the shrinkage of arable land slows gradually.

On a spatial scale, there is a clear difference in the main driving factors of cultivated land change between the Tokyo metropolitan area and Beijing, Shanghai and Guangzhou. Residential area and population are the main driving factors for the former, while economic development and urbanisation are the main driving factors for the latter three areas. This finding is certainly related to socioeconomic development stage. Since 2000, Tokyo has entered a post-industrial society, its primary goal is to pursue efficiency in population agglomeration and higher quality of life. Meanwhile, it has become an important task for the development of urban agriculture to give full play to the multi-functional characteristics of cultivated land resources and provide urban residents with places for tourism, sightseeing and leisure. Agricultural population is gradually decreasing and aging, so the rate of contraction of arable land is slowing down and gradually stabilising, while the arable land per agricultural household even increases slightly. However, Beijing, Shanghai and Guangzhou are still in the late stage of industrialisation and urbanisation, and the coordinated promotion of economic growth, urbanisation and industrial reconstructing are still the main tasks of their development. With the population gathering to the metropolises and the expansion of economic scale, as well as the further improvement of living conditions of urban residents, there is still a strong demand for cultivated land resources, so the contraction of cultivated land will continue for a certain period of time. Over the past two years, Beijing and Shanghai have begun to strictly control the size of their population and construction land, while Guangzhou continues to attract migrants.

On a spatial scale, we also found that there is a rather obvious difference in the main driving factors of cultivated land change among Beijing, Shanghai and Guangzhou. GDP is the primary factor leading to the shrinkage of arable land in Beijing, built-up area is the primary factor in Shanghai, and the Engel's coefficient for rural residents is the primary factor in Guangzhou. In the process of industrialisation and urbanisation, Beijing has focused on expanding the size of its economy to improve its urban competitiveness, which has increased the use of farmland. Shanghai has attached to the expansion of built-up areas to enhance the overall strength of the city, thus increasing the occupation of farmland. Guangzhou has paid attention to the optimisation of consumption structure and industrial structure to promote the city level, thus reducing the occupation of arable land. Even if the urban macroscopic development background and development stages are similar, but due to the objective differences in geographical location, land use scale, population size and density, cultural tradition and life style among the three major cities, not only form obvious heterogeneity of city functions, concentrated forms, ways of expansion and policy orientations, but also result in their real differences in measures for the utilisation and protection of cultivated land.

It is gratifying that under the background of shrinking arable land, metropolitan areas have focused on developing urban characteristic agriculture and have effectively realised the transformation from traditional agriculture to modern agriculture. The primary mission of urban agriculture is not to produce agricultural products, but to develop ecological agriculture, sightseeing agriculture and tourism agriculture that combine with the tertiary industry. For example, Tokyo takes agricultural land as an important space to compensate for the simplicity of urban facilities. In the concept of constructing "the city integrated agriculture", agricultural land is protected as a green open space, and urban agriculture is actively developed. Additionally, agricultural land fully displays the production

and supply function of fresh agricultural products from urban agriculture, the educational function of edifying taste, the landscape function of creating a green living environment, the safety function of disaster prevention and resistance and the enjoyment function of leisure and entertainment. In urban agriculture, advanced science and technology are introduced, vegetable rotation is adopted, greenhouse facilities and factory production are present and the specialisation and expansion of agriculture is improved. More than half the cultivated area is devoted to vegetables, followed by flowers, fruits and seedlings. In addition, urban space is fully utilised through the development of "high-rise fields" and the construction of "underground farms". In 2016, 228,000 households were engaged in urban agriculture in Japan, accounting for 11% of the total number of households in the country. The cultivated land area for urban agriculture is 72,000 hectares, accounting for approximately 2% of the country's total cultivated land area. The output value of urban agricultural operations was 446.6 billion Japanese yen, representing for 8% of the country's total agricultural output. The average annual income of farmers in urban agricultural areas reached 6.1 million Japanese yen, higher than the national average of 4.6 million Japanese yen [50]. Beijing's urban agricultural development is also distinctive. In 2017, there were 1216 agricultural tourism parks in Beijing, receiving 21.053 million person-visits annually and generating a total revenue of 3 billion yuan. Folk tourism generated 22.321 million person-visits, and the total revenue of folk tourism was 1.42 billion yuan [34].

This paper mainly makes a comparative analysis on the driving factors of cultivated land change among East Asian metropolises but does not make a more detailed analysis on the driving factors within each metropolis. Moreover, the selection of driving factors still has the possibility of further expansion and refinement.

## 5. Conclusions and Policy Implications

The shrinkage of arable land is a common feature of land use change in metropolitan areas, while changes in cultivated land per capita or per household are different in different cities. Of course, there are some differences among cities. For example, Tokyo and Guangzhou still have a certain amount of cultivated land in central urban areas, which encompasses 7130 ha (2015) and 971 ha (2017), respectively. However, there is no arable land in the central areas of Beijing and Shanghai. In order to effectively coordinate the contradiction between economic development and resource shortage and environmental protection in metropolitan areas, it is necessary to implement the development strategy of "city integrated agriculture" [51]. As a unique scattered agricultural form, these agricultural lands embedded in the metropolis have long been an organic part of the development of the metropolis, and their irreplaceable multifunctional role has increasingly been accepted by all sectors of society. Therefore, in today's society, where the population is ageing, farmers are part-time workers, and the ecological space is fragmented in metropolitan areas, we should be more respectful of cultivated or agricultural land in urbanised areas of the metropolis.

As a renewable resource, cultivated land can not only provide the growth space for urban and industrial development, but also provide food for present and future generations and support the suitability of landscape environments and natural environmental factors, such as wildlife and water quality [52]. In the suburbs of large cities, maintaining cultivated land for vegetables and fruits is more conducive to meeting the specific food needs of residents of large cities than other agricultural land. Undoubtedly, it is still necessary to ensure that urban residents have more than 50% self-sufficiency in vegetables and more than 90% self-sufficiency in leafy greens, which is the minimum for food security in metropolitan areas.

Additionally, the shrinking of cultivated land in large cities shows that its production function tends to weaken, but its social and ecological functions are increasingly strengthened. This trend in evolution is not a decline in the status and function of cultivated land; on the contrary, it is the diversification of cultivated land functions and the comprehensive promotion of multiple values. From this point of view, strengthening the protection of arable land in metropolitan areas is always necessary.

The socioeconomic factors that affect the contraction of cultivated land are also varied. The research by Wu et al. [38] found that population growth, economic development, urbanisation level improvement, agricultural structure adjustment and urban and rural construction land expansion led to the occupation of a large amount of arable land in Beijing during the 20 years from 1981 to 2001. However, they only performed a qualitative analysis, not a quantitative study. In addition, they focused only on the adjustment of agricultural structure and ignored the more important impact of the development of non-agricultural industries. Gao [39] selected nine indicators for quantitative analysis from three aspects of social, economic and agricultural production and concluded that economic factors (added value of tertiary industry, living expenses of rural residents, total retail sales of consumer goods) were the main factors underlying the decrease in cultivated land in the Shanghai suburbs during the period 2006–2015. However, due to the short time span of her analysis, the selected driving factors were limited to the suburbs, and she failed to comprehensively study the driving factors of farmland shrinkage from the perspective of the overall development of Shanghai; therefore, some of the more important drivers may have been missing. In this paper, industrial restructuring is included in the evaluation index system, mainly because establishing an industrial structure dominated by the modern service industry is the main goal of sustainable development of emerging global cities and agricultural production conditions are not included in the evaluation index system, mainly because they are more the result of urbanisation than the cause.

It should be noted that effective government planning and management and land consolidation policies have led to increases in the total area of cultivated land in Shanghai in recent years (from 2014 to 2018). This indicates that planning and policy factors are also important driving factors for the change of arable land. Unfortunately, however, these factors are often ignored in quantitative analysis because they are difficult to quantify. In a new global agenda entitled "Transforming our World: The 2030 Agenda for Sustainable Development", 17 sustainable development goals and 169 targets are designed [53], which are integrated and indivisible and balance the three dimensions of sustainable development: economic, social and environmental. In the next decades, 95 per cent of urban expansion will take place in the developing world. In order to make cities and human settlements inclusive, safe, resilient and sustainable, it is important and wise to treat arable land in metropolitan areas well. They help create jobs, alleviate the social risks caused by land-lost farmers, build resilience to disasters, and convey benefits from the diversification of urban landscapes. The ultimate goal of urbanisation is not to eliminate farmland, farmers and countryside but rather to comprehensively improve the quality of life and environment of urban and rural residents.

**Author Contributions:** All authors contributed to the work in this paper. Y.S. (Yaxin Shi) and Y.S. (Yishao Shi) designed the research and wrote the paper. Y.S. (Yaxin Shi) participated in calculation of data and the creation of the graphics. All authors have read and agreed to the published version of the manuscript.

**Funding:** This research was supported by the National Key Research and Development Program of China (2017YFA0603102) and the one of key projects for Shanghai General Land Use Planning Revision (2015(D)-002(F)-11).

**Conflicts of Interest:** The authors declare no conflict of interest.

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
