# Peer review of "Spatio-Temporal Variation Characteristics and Driving Forces of Farmland Shrinkage in Four Metropolises in East Asia"

_sustainability, doi:10.3390/su12030754_

Round 1
Reviewer 1 Report
I am attaching an annotated pdf with comments, but here is a summary of the most major issues, as I see them:
1) The paper measures the change of agricultural land within five metropolitan areas, then analyzes the driving influences for four of them. (Seoul is dropped). It might actually be better if you just focus on the three Chinese cities, since presumably the data for them would be more consistent. One problem is that this paper is crying out for some maps. The title suggests there is "spatio-temporal" variation, yet the boundaries of the cities or metro areas are never shown. It seems like this would be pretty important: if you include or exclude exurban or suburban areas could make a big difference in differences in these areas. It felt a bit arbitrary: I would have preferred to see some kind of standardized way of defining what is the analysis area of each city, and being able to see their boundaries on a map, preferably a land use map of some kind.
2) The premise of the paper is that agricultural land within "urban areas" is lost over time as those cities get more urban. This is not really surprising at all: in fact it would be amazing if it did not happen. It seems very common that as cities get more urban, some agricultural land is lost to development. The "driving factors" given are things like population density, but also measures of agricultural output, such as total grain output. Yet is not total grain output reduction a RESULT of loss of agricultural land, as opposed to causing it? Seems like that would be analogous to. let's say if I can run 10 km in 40 minutes when I am young, but only run it in 60 minutes when I am much older, saying that the "decrease in 10 km run time caused aging". It is not a cause of the aging but rather a result. There are a number of "driving forces" variables in Table 7 which are like this. Most of them, I would say. In my opinion this fatally flaws the paper.
Indeed why agricultural land might be lost is a highly economic decision. Things like land prices, distance to market, commodity prices, cost of transportation. global trends, all seem like would be important things to look at. What was done here feels way too simplistic. Sorry.
3) The organization of the paper should be redone into more classical sections of Intro, Methods, Results, Discussion, etc. As it stands now there is a mixed methods and results in sections 3 and 4.
4) Baidupedia not a good citation source. Wikipedia is typically not accepted as a valid citation source in scientific papers.

Author Response
Dear Review 1,
Thank you very much for all your comments and suggestions. Some questions are answered as follows.
I am attaching an annotated pdf with comments, but here is a summary of the most major issues, as I see them:
Response:We have responded to each of your comments and have supplemented or modified our manuscript.
The paper measures the change of agricultural land within five metropolitan areas, then analyzes the driving influences for four of them. (Seoul is dropped). It might actually be better if you just focus on the three Chinese cities, since presumably the data for them would be more consistent. One problem is that this paper is crying out for some maps. The title suggests there is "spatio-temporal" variation, yet the boundaries of the cities or metro areas are never shown. It seems like this would be pretty important: if you include or exclude exurban or suburban areas could make a big difference in differences in these areas. It felt a bit arbitrary: I would have preferred to see some kind of standardized way of defining what is the analysis area of each city, and being able to see their boundaries on a map, preferably a land use map of some kind.
Response:Okay. We have added some maps. Please see the revised manuscript.
The premise of the paper is that agricultural land within "urban areas" is lost over time as those cities get more urban. This is not really surprising at all: in fact it would be amazing if it did not happen. It seems very common that as cities get more urban, some agricultural land is lost to development. The "driving factors" given are things like population density, but also measures of agricultural output, such as total grain output. Yet is not total grain output reduction a RESULT of loss of agricultural land, as opposed to causing it? Seems like that would be analogous to. let's say if I can run 10 km in 40 minutes when I am young, but only run it in 60 minutes when I am much older, saying that the "decrease in 10 km run time caused aging". It is not a cause of the aging but rather a result. There are a number of "driving forces" variables in Table 7 which are like this. Most of them, I would say. In my opinion this fatally flaws the paper. Indeed why agricultural land might be lost is a highly economic decision. Things like land prices, distance to market, commodity prices, cost of transportation. global trends, all seem like would be important things to look at. What was done here feels way too simplistic. Sorry.
Response:Okay. According to your advice, we have changed some indicators in the original Table 7 (now Table 2) and Table 8 (now Table 3), and the calculation and analysis are carried out again.
The organization of the paper should be redone into more classical sections of Intro, Methods, Results, Discussion, etc. As it stands now there is a mixed methods and results in sections 3 and 4.
Response:Okay. We have reconstructed the paper according to the classical order: Intro, Data and Methods, Results, Discussion, Conclusions and Policy Implications.
Baidupedia not a good citation source. Wikipedia is typically not accepted as a valid citation source in scientific papers.
Response:Okay. We have found another source.
Please see the attachment.

Reviewer 2 Report
The paper entitled Spatio-temporal variation characteristics and driving forces of farmland shrinkage in five metropolises in East Asia” presents and in-depth assessment of the driving forces of land take in megalopolis using correlation, principal component analysis and regression to verify the observed trends.
I found the paper overall well-written and easy-to-understand which is crucial while commenting this field of research.
Also the methodology: is quite interesting and easy-to-replicate, which is a crucial issue for this Journal.
I just have a couple of remarks that regards the structure:
please re-structure your chapters reaching a clear separation between introduction, methodology, results and discussion and the conclusions; be more detailed while describing your methods (also in the abstract) and the utilization of sources (it is not clear if the description if sites and the data collection is part of your own methodology or is introductive to further analysis); in the discussion, please refer only to your work and not refer to external works. Or you can do it to a certain extent in the conclusive remarks: enforce the policy implications of your findings.See my few detailed comments in the attached file.
Good luck!

Author Response
Dear Review 2,
Thank you very much for all your comments and suggestions. Some questions are answered as follows.
I just have a couple of remarks that regards the structure:
please re-structure your chapters reaching a clear separation between introduction, methodology, results and discussion and the conclusions; be more detailed while describing your methods (also in the abstract) and the utilization of sources (it is not clear if the description if sites and the data collection is part of your own methodology or is introductive to further analysis); in the discussion, please refer only to your work and not refer to external works. Or you can do it to a certain extent in the conclusive remarks: enforce the policy implications of your findings.
Response:Okay. We have re-structured the chapters of our manuscript and made the corresponding modification. Please see the revised manuscript.
Please see the attachment.
See my few detailed comments in the attached file.Response:We have responded to each of your comments and have supplemented or modified our manuscript.

Reviewer 3 Report
The manuscript “Spatio-temporal variation characteristics and driving forces of farmland shrinkage in five metropolises in East Asia” is an interesting paper that focusing on the shrinkage of cultivated land in different metropolitan areas (Tokyo, Seoul, Beijing, Shangai and Guanzhou). The topic fits the aims of the journal and is relevant on the discourse of sustainable planning of contemporary urban contexts. In my opinion, this is an interesting topic for academic readers. I think that the authors have rich material and that’s why I would assign for this paper MINOR REVISIONS. See below for the others detailed remarks:
Since data is missing for Seoul metropolitan area, the authors should both modify the title (four metropolises in East Asia) and remove everything related to Seoul.
Introduction:
Lines 36-41: rephrase the sentence, difficult to follow.
Furthermore, The authors should better specify the main objective of their work.
Before the section 2 (Overview of the five metropolises in East Asia) the authors should insert the Data and methods sections. This is necessary for improve the readability of the paper. This new section will contain paragraph 4.1.1., 4.2.1 and some paragraph contents of the 4.1.2. and 4.2.2.. They should describe the used methods and in the results paragraph their findings. Furthermore, I suggest to the authors to include a flowchart which can facilitate the reader in understanding this part. The flowchart can increase the clarity of the various phases that compose the methodology and at the same time synthesize of the source of data adopted in the case study.
It is necessary that the authors insert a separated results paragraph.
Overview of the five metropolises in East Asia
Table 1: Replace notes with Data reference year.
Spatio-temporal variation in cultivated land shrinkage in five metropolises in East Asia
In my opinion, this paragraph should be moved to the results.
Table 2: Use hectares as a unit (also for the table 3).
The authors just enter the numbers of cultivated area, agricoltural population and per capita arable land area for the five metropolises (sections 3.1,3.2,3.3,3.4,3.5) without delving into the reasons for these increases / decreases. They should improve this part (see lines 370 – 384).
Furthermore, they use a multiple crop index for describe the utilization rate of arable land but the formula is not specified neither the references.
Line 139: The decrease in cultivated land was 2,023 hectares not 1,361 hectares.
Lines 176 – 180: The authors must specify that this sentences refer to Beijing.
Conclusions:
Make it at least half shorter. State only, what is new, how they contributed to science and if there can anything else evolve from your findings. Furthermore, the authors should add some comments about their findings contribute to the aims of the 2030 Agenda, and in particular goal 11 (11.b): “Make cities and human settlements inclusive, safe, resilient and sustainable”.
Author Response
Dear Review 3,
Thank you very much for all your comments and suggestions. Some questions are answered as follows.
Since data is missing for Seoul metropolitan area, the authors should both modify the title (four metropolises in East Asia) and remove Seoul.
Response:Okay. We have removed Seoul.
Introduction: Lines 36-41: rephrase the sentence, difficult to follow.
Furthermore, The authors should better specify the main objective of their work.
Response:Okay. We have revised it.
Before the section 2 (Overview of the five metropolises in East Asia) the authors should insert the Data and methods sections. This is necessary for improve the readability of the paper. This new section will contain paragraph 4.1.1., 4.2.1 and some paragraph contents of the 4.1.2. and 4.2.2.. They should describe the used methods and in the results paragraph their findings. Furthermore, I suggest to the authors to include a flowchart which can facilitate the reader in understanding this part. The flowchart can increase the clarity of the various phases that compose the methodology and at the same time synthesize of the source of data adopted in the case study.
Response:Okay. We have reconstructed it. Please see the revised manuscript.
It is necessary that the authors insert a separated results paragraph.
Response:Okay. We have added Results section.
Overview of the five metropolises in East Asia
Table 1: Replace notes with Data reference year.
Response:Okay. We have revised it.
Spatio-temporal variation in cultivated land shrinkage in five metropolises in East Asia
(1)In my opinion, this paragraph should be moved to the results.
Response:Okay.
(2)Table 2: Use hectares as a unit (also for the table 3).
Response:hm2 is also hectares.
The authors just enter the numbers of cultivated area, agricultural population and per capita arable land area for the five metropolises (sections 3.1,3.2,3.3,3.4,3.5) without delving into the reasons for these increases / decreases. They should improve this part (see lines 370 – 384).
Response:Okay. We have improved this part. Please see the revised manuscript.
Furthermore, they use a multiple crop index for describe the utilization rate of arable land but the formula is not specified neither the references.
Response:Okay. We have added the definition and references.
Line 139: The decrease in cultivated land was 2,023 hectares not 1,361 hectares.
Response:No. It is the Seoul metropolitan area, not Seoul city; and it is the average annual decrease in cultivated land area, not the total decrease in cultivated land area. But we have removed Seoul.
Lines 176 – 180: The authors must specify that this sentences refer to Beijing.
Response:Okay. We have added it.
Conclusions:
Make it at least half shorter. State only, what is new, how they contributed to science and if there can anything else evolve from your findings. Furthermore, the authors should add some comments about their findings contribute to the aims of the 2030 Agenda, and in particular goal 11 (11.b): “Make cities and human settlements inclusive, safe, resilient and sustainable”.
Response:Okay. We have revised the Conclusions and Policy Implications section. Please see the revised manuscript.
Round 2
Reviewer 1 Report
The major problem: it still seems like measures of agricultural output are given as "causal factors". For example in Tokyo agricultural production is given as the driving factor causing decrease in cultivated lands (line 245). In Beijing rural agricultural production conditions and rural electricity consumption is given (lines 452 and 470). These again seem to me to be the RESULT of urbanization, not the cause of it. If farmland is replaced by a housing area with 10,000 residents it seems obvious that electricity consumption would increase there. That didn't cause the loss of the farmland, it is the result of it. This again seems to me to be a completely incorrect way to analyze the loss of agricultural lands.
Author Response
Response to Reviewer 1
Dear Review 1,
Thank you very much for your comments and suggestions once again. Your questions are answered as follows.
The major problem: it still seems like measures of agricultural output are given as "causal factors". For example in Tokyo agricultural production is given as the driving factor causing decrease in cultivated lands (line 245). In Beijing rural agricultural production conditions and rural electricity consumption is given (lines 452 and 470). These again seem to me to be the RESULT of urbanization, not the cause of it. If farmland is replaced by a housing area with 10,000 residents it seems obvious that electricity consumption would increase there. That didn't cause the loss of the farmland, it is the result of it. This again seems to me to be a completely incorrect way to analyze the loss of agricultural lands.
Response:According to your advice, we have deleted “agricultural production conditions” factor, and revised the manuscript again.